# A Systematic Review on Modeling Methods and Influential Factors for Mapping Dengue-Related Risk in Urban Settings

**DOI:** 10.3390/ijerph192215265

**Published:** 2022-11-18

**Authors:** Shi Yin, Chao Ren, Yuan Shi, Junyi Hua, Hsiang-Yu Yuan, Lin-Wei Tian

**Affiliations:** 1Faculty of Architecture, The University of Hong Kong, Pokfulam, Hong Kong SAR, China; 2School of Architecture, South China University of Technology, Guangzhou 510641, China; 3Department of Geography and Planning, University of Liverpool, Liverpool L69 3BX, UK; 4School of International Affairs and Public Administration, Ocean University of China, Qingdao 266100, China; 5Department of Biomedical Sciences, City University of Hong Kong, Kowloon, Hong Kong SAR, China; 6School of Public Health, LKS Faculty of Medicine, The University of Hong Kong, Pokfulam, Hong Kong SAR, China

**Keywords:** dengue, risk mapping, urban environment, influential factors, spatial models

## Abstract

Dengue fever is an acute mosquito-borne disease that mostly spreads within urban or semi-urban areas in warm climate zones. The dengue-related risk map is one of the most practical tools for executing effective control policies, breaking the transmission chain, and preventing disease outbreaks. Mapping risk at a small scale, such as at an urban level, can demonstrate the spatial heterogeneities in complicated built environments. This review aims to summarize state-of-the-art modeling methods and influential factors in mapping dengue fever risk in urban settings. Data were manually extracted from five major academic search databases following a set of querying and selection criteria, and a total of 28 studies were analyzed. Twenty of the selected papers investigated the spatial pattern of dengue risk by epidemic data, whereas the remaining eight papers developed an entomological risk map as a proxy for potential dengue burden in cities or agglomerated urban regions. The key findings included: (1) Big data sources and emerging data-mining techniques are innovatively employed for detecting hot spots of dengue-related burden in the urban context; (2) Bayesian approaches and machine learning algorithms have become more popular as spatial modeling tools for predicting the distribution of dengue incidence and mosquito presence; (3) Climatic and built environmental variables are the most common factors in making predictions, though the effects of these factors vary with the mosquito species; (4) Socio-economic data may be a better representation of the huge heterogeneity of risk or vulnerability spatial distribution on an urban scale. In conclusion, for spatially assessing dengue-related risk in an urban context, data availability and the purpose for mapping determine the analytical approaches and modeling methods used. To enhance the reliabilities of predictive models, sufficient data about dengue serotyping, socio-economic status, and spatial connectivity may be more important for mapping dengue-related risk in urban settings for future studies.

## 1. Introduction

Dengue fever (DF) is one of the most prevalent mosquito-borne viral diseases in the world and causes a huge health burden worldwide. The lack of effective mosquito control, changing lifestyles, unplanned urbanization, and globalization are cited as the major drivers of the spread of dengue [1]. Female *Aedes* species mosquitoes are the main vector of dengue virus (DENV), namely *Aedes aegypti* and *Aedes albopictus* [2]. Both are highly domesticated urban mosquitoes and prefer to live with humans in and around their homes, feed on humans, and lay eggs in small containers. The process of rapid urbanization will likely generate high demographic regions with disproportionate infrastructure quality, which creates ideal conditions for mosquito breeding and increases the risk of DENV transmission. In the last two decades, the number of DF cases has increased sharply to over 5.2 million in 2019 [3]. Meanwhile, climate change and the urban heat island phenomenon result in a warmer environment in urban areas, favoring the proliferation of both vectors and dengue virus [4]. According to a prediction by Messina et al. [5], 2.25 billion more people might be faced with a risk of DF at the end of the 21st century, compared with 2015. By that time, some megacity aggregated regions with no DF risk areas could become suitable for the spread of dengue virus, such as the southeastern USA, coastal eastern China, Japan, and inland areas of Australia.

To break the chain of DENV transmission and reduce the DF incidence rate, an incidence or risk map is the one of most effective tools for public health management. The resultant maps not only intuitively illustrate the spatial or spatio-temporal pattern of dengue-related risk, but also propose determinants based on spatial modelling [6]. Though there is a rich literature about developing reliable mathematic models for displaying the interaction between dengue transmission and influential factors, most of them ineffectively predict or evaluate the risk in the region without any sampling or surveillance data. In the meantime, an abstract scientific model is difficult for use by non-specialists. Therefore, taking the advantage of visualization and prediction, mapping dengue-related risk is valuable for local governments and stakeholders to detect the locations of hot spots and take action in controlling disease transmission.

Many review studies have investigated influential factors on DENV transmission, including the impacts of socio-economic factors [7], urban landscape [8], and climatic variables [9]. Various modelling techniques have also been discussed, such as quantitative approaches [10] and Bayesian methods [11]. However, mapping technologies have been less structurally reviewed in the last decades. So far, Louis et al. [12] is the only review on the available mapping tools, based on papers published until 2014. Four types of mapping are defined in that study, namely descriptive, validated, predictive, and early warning system (EWS) maps. The descriptive map is based on clustering analysis rather than predicting models, whereas the EWS attempts to establish criteria in early recognition of disease outbreaks for applications in public health. Only 9 of 26 papers generated predictive or EWS for DF risk in Louis et al. [12].

With the evolution of predictive maps for dengue-related risk in recent years, the spatial patterns of DF risk can be predicted at different scales, such as on the global [5], continental [13], national [14,15,16,17], and urban level [8,18]. However, mapping dengue-related risk at a large scale will significantly underrepresent dengue occurrence in urban regions [19]. First, as interpreted in Attaway et al. [20], mapping approaches in urban and rural areas should be different because the data availability and characteristics of these regions are largely distinct. Secondly, mapping of dengue endemicity on the global scale is usually coarse because global climate datasets have relatively low spatial resolution [21]. Lastly, a model developed with site-specific data is often restricted within a small region that shares a similar climate, geography, urbanization, life behaviors, and so on. Therefore, controversial outcomes and significant uncertainty may be resulted when applying a developed model on predicting the spatial pattern of dengue-related risk at a larger scale.

Currently, there have been few review studies focused on the predictive maps of dengue-related risk in urban settings. Investigating the risk at a relatively small scale, such as in a single city or urban agglomeration, may be more valuable for local interested administrations to guide urban planning and implementation of policies for preventing disease spread, as this could capture the complexities that contribute to a risk distribution, which may be missed at a larger scale. Thus, the objectives of this research include:
(1)Reviewing available modeling tools for generating predictive maps of dengue-related risk since 2014; (2)Investigating determinants in urban settings used for spatial and spatio-temporal modeling; (3)Discussing the limitations and advantages of different methods for developing dengue-related risk; (4)Proposing improvements for future works.

## 2. Methods

The review was conducted in accordance with the Preferred Reporting Items for Systematic Reviews and Meta-Analyses (PRISMA) statement. We developed a systematic review protocol and registered with PROSPERO. All the review team members followed the protocol established at the beginning of the review.

### 2.1. Search Terms and Selection Criteria

Five major academic search databases were accessed to retrieve the relevant studies, namely Science Direct, ProQuest, Web of Science, PubMed, and Scopus. As shown in Table 1, a set of four-level querying was conducted in each database, respectively.

As Louis et al. [12] reviewed the mapping tools before 2013, this research only focused on articles published after January 2014. In that previous review study, only the maps modelled with dengue fever cases were included. However, Sallam et al. [22] found that mapping entomological risk, such as mosquito presence or abundance, could also be a proxy for transmission risk of various vector-borne diseases, especially in non-dengue epidemic regions that still have potential dengue risk from future changes in climate and importation from abroad. Therefore, both epidemic and entomological risk mapping for the dengue virus were included in this research. Meanwhile, descriptive maps were excluded as no predictive model was developed. Finally, the study complied to all 5 inclusion criteria to determine selection for further analysis:Articles published from January 2014 to May 2022;In an English peer-reviewed paper or non-conference proceedings;A spatial or spatio-temporal modeling tool employed to predict the potential dengue risk distribution;A predictive risk map or early warning system about dengue-related burden in a city or area of urban agglomeration as the outcome;Includes an investigation of the impacts of environmental and socio-economic determinants for modeling.

### 2.2. Data Extraction

All data were manually extracted from selected studies and categorized into four strands: general observation, data characters, modeling approaches, and influential factors. In the first section, the location and basic information about the investigated urban area were collected. Then, information about the spatial resolution of the outcome map, surveillance data applied in studies, mosquito species, and buffers were recorded. The variables applied in modeling were further retrieved and grouped into climate, built environment, socio-economic, and others, according to their natures. Additionally, there were 7 and 4 subgroups in the built environment and socio-economic groups, respectively, due to the number of variables in these two groups being very large. The data sources and corresponding temporal resolutions of climatic data were recorded, i.e., from monitoring records or remote sensing. Thirdly, the analytic methods, including spatial and temporal models and spatial connectivity assumptions were collected and classified according to their functions and purposes. Spatial connectivity assumptions were grouped into three categories, i.e., distance-based, human movement, and vector movement, accordingly [23]. In the last section, the influential factors in each outcome model and corresponding effect were compared. If the factor was included in the final predicting model but no coefficient was reported in the study, their effects were marked as unknown. Finally, the applications of these incidence maps, limitations, and future works were proposed.

## 3. Results

### 3.1. General Observation

As shown in Figure 1, in total 518 non-repetitive articles were extracted from the five databases. Based on the above inclusion criteria, 220 articles were chosen for abstract review after reading their titles, and then 68 papers entered the round for full-text review. Finally, a total of 28 qualified articles were selected and numbered in chronological order (listed in Table 2). Twenty papers directly investigated the risk of dengue incidence and were designated by a combination of “D” and corresponding numbers in the following sections, e.g., D1 is a proxy for the article of Dickin and Schuster-Wallace [24]. Eight of the selected papers developed mosquito presence or abundance as a proxy for the risk of dengue outbreaks, and “A” was combined with the corresponding number of articles, e.g., A1 is a proxy for Machault et al. [25].

The information about the location, year, and numbers of selected articles are overlaid with a map of global dengue burden and 10 °C isotherms in the winter season (Figure 2). Almost all the selected references were located in the “Frequent or Continuous” region in Southeast Asia and South America. Three studies investigated potential dengue risk in a no evidence region in North America (A4 and A6) and Oceania (A8). The dengue burden was investigated by multiple scholars in Pearl River Delta (four papers), Southern Taiwan (three papers), West Bengal in India (two papers), and Java in Indonesia (two papers). As shown in Figure 3, the population densities of those urban areas are predominantly in the ranges of 100 to 10,000 inhabitants per km^2^. The densest environment investigated was Kolkata, India (almost 22,000 inhabitants per km^2^). The Tartane possessed the sparsest inhabitants among all investigated areas.

### 3.2. Data Characters

As shown in Table A1, most studies predicted dengue risk at a district level or in grids with few hundreds of meters. A1 even evaluated the entomological dengue risk at the building level. 

Only two studies predicted dengue-related risk without any surveillance data, namely D10 and A8. A8 modeled the mosquito population instead of surveillance data from mosquito traps. In terms of species, *Aedes aegypti* was the focus of twelve papers, whereas *Aedes albopictus* was only investigated in five studies. Another vector of DENV, *Culex,* was studied in A7. Meanwhile, six studies simultaneously mapped the two species of *Aedes* mosquito. Considering the limited flying capacity of mosquitoes, a buffer zone was employed in eight studies and the radius of the mosquito trap ranged from 50 to 500 m. 

Regarding environmental and socio-economic variables for predicting dengue-related risk, the climatic variables were the most popular predictors. Half of them had monthly temporal resolution, followed by yearly, seasonally, and daily. In addition, seven studies used remote sensing data rather than weather monitoring datasets. In terms of built environmental variables, land use and land cover (LULC), landscape, and road were the most common type of factors for modeling. Population density was the most common socio-economic variable for assessing dengue risk, followed by demographic information. 

It is worth noting that there were four special variables for mapping dengue risk in selected studies. Cellphone data was used for evaluating the mobility of humans in D3; D18 employed future socio-economic development data to predict potential dengue risk in scenarios of urban expansion from 2015 to 2050. D19 analyzed pictures from the Google Street Views dataset and extracted eight types of artificial container densities as predictors; A8 considered the effect of non-compliant rainwater tanks increasing entomological dengue risk.

### 3.3. Modeling Approaches

As shown in Figure 4, the analytic methods employed in selected studies were categorized into four main sections, including cluster analysis, covariates screen, spatial (temporal) modeling, and calibration or validation. 

#### 3.3.1. Cluster Analysis

Moran’s I and kernel density were the most popular techniques for detecting spatial patterns and hot spots according to epidemic or entomological surveillance data in preprocessing. D13 used scan statistics (ScSAT) for assessing probability risk, and then applied local Moran’s I for cluster analysis. D17 combined kernel density and Getis-Ord Gi* for detecting the hot spot of dengue burden. 

#### 3.3.2. Covariates Screen

Over half of the selected studies employed correlation coefficients (r), principal component analysis (PCA), probability value (*p*-value), or variance inflation factor (VIF) for reducing multi-collinearity between covariates. Pearson’s r was employed in seven studies, and only one applied the Spearman’s r. Some studies used multiple methods to select variables, such as VIF and *p*-values, which were adopted in D20.

#### 3.3.3. Spatial Modelling

There were five classes of spatial or spatio-temporal modeling employed in selected studies, namely weighting, statistics, linear regression, Bayesian models, and machine learning. Only four studies applied a weighting method. The geographical weighting regression (GWR) and MaxEnt model were employed in three studies for mapping epidemic dengue risk. In terms of entomological risk, the MaxEnt model and generalized linear model (GLM) with negative binomial distribution are dominant models. There are four kinds of Bayesian models recorded, namely Bayesian maximum entropy, normal Bayesian models, conditional autoregressive (CAR) or Leroux CAR, and Markov random fields. Regarding the outcome of mapping, only D4, D5, and D20 developed the early warning system (EWS) for dengue outbreaks by ScSAT or Bayesian models. 

In addition, almost all studies applied distance-based spatial connectivity assumptions in modeling, except in D3 and A8. D3 employed cellphone tracking data cooperating with blood meal hunting behavior of *Aedes albopictus* to assess the probability of acquiring the DF virus as the indicator of importation risk. A8 was the only study that used a mechanistic model. An agent-based model was adopted to simulate the spread of mosquito population as the key predictor for GLM to evaluate the potential invasion of dengue fever mosquitoes.

#### 3.3.4. Calibration and Validations

Almost all studies applied various statistical metrics to evaluate the predictive performance according to the type of model, except D10. Wen et al. [27] and Desjardins et al. [34] employed the Akaike information criterion (AIC) and deviance information criterion (DIC) to select the best-fitted model. For the weighting method and linear regression, the r, R-squared, and standard deviation of the residuals, namely RMSE and MAE, were the main indices. Almost all machine learning models applied the area under the receiver operating characteristic (AUC) or weighted Kappa (Kw) to evaluate their outcome models. Additionally, some studies applied cross-validation for r, measuring the performance of a predictive model, such as leave-one-out and k-fold cross-validation [28,33,35,38]. 

### 3.4. Influential Factors

#### 3.4.1. Climatic Variables

As shown in Table 3, air temperature, LST, and precipitation were the most common determinants in modeling dengue-related risk. There were nine and seven studies that developed their final model by air temperature and LST, respectively. However, their effects on dengue-related risk were not consistent with each other. Surprisingly, only D20 found the temperature positively related to dengue risk, whereas such correlations in D7 and D18 were nonlinear. Similarly, the effects of LST on dengue risk were positive in three studies and negative in four studies. In terms of entomological risk, A3 and A4 reported positive relationships between air temperature and mosquito abundance. However, A6 found that the relation was nonlinear. Regarding precipitation, 11 studies applied this variable in developing their models. Mosquito abundance was positively associated with rainfall in all studies about entomological risk, except it was nonlinear in A3. However, only three studies about dengue risk reported the positive impact of precipitation, whereas D14 and D18 indicated negative correlations. Other variables were less employed in modeling and demonstrated few controversies, except in the daily temperature range (DTR), which positively correlated with dengue risk in D14 but negatively in A3. In addition, the lag effect of meteorological data on DF risk or mosquito abundance was only considered in D14.

#### 3.4.2. Built Environmental Variables

Built-up/impervious areas, normalized difference vegetation index or enhanced vegetation index (NDVI or EVI), greening area, road density, proximity to water, and establishments were the most popular variables in modeling dengue risk. Especially for built-up or impervious areas, seven studies reported its positive association with both epidemic and entomological risk (Table 4). Only A1 and A6 indicated non-positive effects. Relevant variables about built-up areas such as residential use, road density, and proximity to specific establishments, including tire shop and cemeteries, increased dengue risk in an urban area. 

The effects of vegetation and water were more complicated. Four studies found that dengue risk was negatively correlated with NDVI, whereas a positive relation was reported in two studies investigating entomological risk and one for epidemic risk. The effects of greening area were only investigated in the studies modeling entomological risk. Greening area presented either positive or nonlinear correlations with entomological risk. Variables about water, such as the normalized difference water index (NDWI) and proximity to water bodies, demonstrated similar conflicting results.

Urban areas with insufficient infrastructures could be located by factors such as vacant use, unplanned areas, and specific facilities such as containers, water tanks, and drain networks. The above variables were positively associated with both epidemic and entomological dengue risk. However, the effects of artificial container types on dengue risk varied with types, according to the research in D19.

Regarding other determinants, the property size on an urban block was the only variable referring to urban morphology applied in selected studies and was negatively associated with mosquito abundance. Three variables of topography were investigated, including flow accumulation, elevation, and slope. The elevation demonstrated a nonlinear correlation with dengue-related risk in two studies and was positive in D13. 

#### 3.4.3. Socio-Economic Variables

As shown in Table 5, variables about the population, namely population and household density, were dominant in developing dengue-related risk models. These two variables mainly demonstrated a positive influence on epidemic and entomological risk. However, D14 found that population density was negatively related to dengue risk, whereas D7 and A6 indicated the relationships were nonlinear. In general, urban areas with high GDP, elders, schooling, and unemployed people were explored for dengue risk. Surprisingly, D6 and D14 reported higher education levels and income demonstrated a greater possibility for infection. In addition, districts with infection history, low neighborhood quality, and no piped water increased the risk of dengue outbreaks and mosquito breeding. However, A6 found that the impact of vacant housing on mosquito abundance was nonlinear.

## 4. Discussion

Compared with the reviewed papers in Louis et al. [12], our review about predictive maps for DF risk in urban settings demonstrated significant evaluations in recent years, including widely expanded study areas, applications of emerging big data and data-mining techniques, and novel modelling approaches.

### 4.1. Study Areas

The dengue fever risk in the cities of India, Nepal, and Pakistan, which are in the frequent or continuous dengue risk level were investigated (Figure 2); there were no studies on these areas reviewed by Louis et al. [12]. However, many areas in Africa demonstrating high possibility for dengue outbreaks are still lacking. Even though Attaway et al. [53] developed a valuable map for dengue suitability for the entire African continent with a fine spatial resolution, there have been almost no detailed studies on any city or urban agglomerated zone in Africa. 

In addition, three studies, i.e., A4, A6, and A8, investigated no evidence risk zones to present the spatial entomological risk as a precaution for vector-borne diseases. A8, which predicted scenarios of *Aedes* mosquito invasion in Brisbane (Australia) rather than modeling by surveillance data such as in A4 and A6, is an especially innovative example that could be applied to other cities located in suitable zones for future mosquito risk. In Louis et al. [12], only one case studied a no evidence risk zone, namely Hu et al. [54], in which the spatial pattern of dengue fever transmission in Queensland (Australia) was investigated.

### 4.2. Effective Predictors

#### 4.2.1. Entomological Data

A significant improvement in data application was found for aspects of entomological data. As reported in Louis et al. [12], though the house-, container- or Breteau-indices were commonly used, the association between these entomological indicators and dengue cases were often not detected [55], due to the low efficiency of the dengue vector sampling method. In recent years, with the wide utilization of adult mosquito captures, such as Ovitraps (D2) and Gravidtraps (D11), entomological data has played a primary role in modeling the spatial distribution of dengue incident ratios. Surveillance data about adult mosquito abundance can be employed for assessing entomological dengue fever risk as well, such as in A4 and A6.

#### 4.2.2. Climatic Data

Louis et al. [12] addressed the sparse distribution of meteorological observations that hindered the spatial resolution of the final generated maps. Though the remote sensing data used as a proxy for temperature, such as the LST or LSTn, had high spatial resolution, their influences on dengue risk were feeble [28,39,48]. This may be due to fewer temporal characteristics and lower accuracy of remote sensing data compared with meteorological records. In addition, both types of climatic data demonstrated a complicated impact on DF risk. Though warm and wet conditions with moderate rainfall are favored by both DF outbreaks and mosquito breeding, some studies found that air temperature and rainfall were negatively correlated with DF risk. As discussed in D14, a frequent short-lived rain before a dengue epidemic destroyed the microhabitats of mosquitoes by washing eggs and larvae of vectors away. Therefore, many studies reported that the effects of temperature and precipitation were nonlinear [41,42,46]. To improve the performance of the model, a set of climatic indicators to describe the climate pattern in a region might be much better than employing individual meteorological variables [34].

#### 4.2.3. Built Environmental Data

The built environmental data for mapping dengue fever risk in urban settings were mainly based on remotely sensed data, such as land cover, land use, landscape, topography, etc. The controversial variables mainly referred to vegetation and water. The complex impact of vegetation may have resulted from different preferences between *Aedes* species. As an endophilic species, the *Aedes aegypti*, investigated in D9, D11, and D13, is more abundant in compact districts with high building coverage and less greening. However, the *Aedes albopictus* is fond of environments with more vegetation to offer them sufficient shelter, shade, and humid outdoor places to feed or breed [56]. Many previous studies found *Aedes albopictus* nonlinearly correlated with vegetation [41,57]. In terms of water bodies, mosquito larvae were not found on open surfaces of large bodies of deep freshwater (e.g., lakes, ponds, rivers, or reservoirs), but were concentrated along the shallow edges, such as most waterfront in urban spaces. Thus, the status of water bodies determined the probability of mosquito presence and dengue virus (DENV) transmission [58].

Except for the conventional remote sensing and administrative data, the street view images dataset from Google was first utilized in D19 for detecting artificial container types in the outdoor environment. As female *Aedes* mosquitoes feed on the blood of their hosts (e.g., humans and other mammals) and breed in any containers with standing water, the lifestyle of residents impacts entomological dengue risk [2]. The data from the street view images may be an important supplement for data from satellite imagery to display the microhabitat environment of mosquitoes. However, some inaccessible zones would be missing data for evaluation, as street view images are only captured along the road [49].

#### 4.2.4. Socio-Economic Data

As much of the socio-economic data applied in mapping DF risk in urban environments were collected from the local census, the spatial resolution of the generated maps were often related to the basic census unit, such as neighborhoods, wards, or districts. The conflicting results regarding the effect of population density on DF risk may relate to the different *Aedes* species as well. The *Aedes aegypti* prefers high population density, which results in a high risk of DENV transmission, whereas *Aedes albopictus* is the opposite and shows nonlinear association. Though many studies reported both children and elders were vulnerable to environmental disasters, school-aged teenagers and young adults were highly exposed to dengue transmission because they had to visit school buses, workplaces, or other districts with high DF risk [38,59,60].

One research gap mentioned in Louis et al. [12] was that the impact of human mobility and movements on dengue transmission risk could not be captured well using the demographic data, which are commonly applied as a proxy for population mobility. Even though the variables about roads and public establishments were positively associated with dengue risk, as shown in Wen et al. [27] and Ong et al. [26], they only represent the mobility in urban zones indirectly. Tracking individual mobility patterns using cell phone data is an effective dataset for evaluating the human movement. In D3, the local transmission risk was represented by the visited time of every cellphone tower, which is more accurate for evaluating human movement and detecting hot spots for dengue burden. 

### 4.3. Modeling Techniques

#### 4.3.1. Predictive Models

Similar to the results in Louis et al. [12], our findings demonstrated that the most popular modelling algorithms for developing a predictive map were the MaxEnt model and GLM. At the same time, other novel modelling methods, including machine learning, GWR, and Bayesian approaches, have been extensively employed for both epidemic and entomological dengue risk mapping in recent years. 

The different specialties between models are significant. All the machine learning methods applied in reviewed studies were supervised learning algorithms, which requires a large amount of data for training the final model. Though machine learning methods achieved the most accurate predictions by developing complex models with large variables, the internal association between predictors may not be clearly demonstrated as in a Bayesian model [61]. As an extension of the traditional multiple linear regression, GWR was often used for examining the potential spatial autocorrelation and dealing with non-stationary data [62]. However, the explanatory power of GWR was limited to estimating the spatio-temporal relationship between associations of incidence and predictors as it was not suitable for making temporal inferences [34]. In contrast, Bayesian models could estimate the relationship between incidence and predictors spatially and temporally based on locally weighted regressions in both geographic and attribute spaces. The CAR was the most popular Bayesian model for predicting dengue risk, according to a review by Aswi et al. [11]. Unlike machine learning methods, Bayesian models can minimize the variance of the estimators, especially in places where the population is small [63]. In addition, Bayesian approaches can predict the lag effect of climate. Therefore, EWS was mainly developed using Bayesian models, such as in D4 and D20. However, Bayesian approaches require highly skilled users with advanced statistics, as this method is too complicated and the spatial resolutions are often limited to districts or due to the large scale [64].

#### 4.3.2. Calibration and Validation

Calibration and validation are the fundamental steps toward creating reliable predictive maps. Our findings presented multiple statistical metrics applied for calibration or validation. However, some studies used a single metric as the absolute performance measure, such as in D7, D13, D18, A2, and A6, which may result in serious drawbacks. For instance, the AUC is one of the most commonly used statistics to assess model performance developed by machine learning algorithms [65]. The AUC interprets probability that a presence cell has a higher predicted value than a pseudo-absence, but ignores the different weights in commission and omission errors. To eliminate possible deviations of the AUC value, using more than one accuracy measure to seek the reliability of the MaxEnt model is necessary [66]. 

### 4.4. Mapping Methodology Design

Though mapping approaches differed in reviewed articles, the design of mapping methods and selection of modeling techniques were generally related to data availability and application purposes. Data, especially from the surveillance of dengue cases, played a big role in both the modelling algorithm and spatial resolution. In cases with sufficient surveillance and geographic data, DF risk could be detected at fine spatial and temporal scales by data-driven machine learning algorithms, such as A1 that successfully detected the entomological dengue risk for each building. In contrast, D16 could only predict dengue burden at the township level in the Mekong delta region, Vietnam, due to a lack of detailed geolocation information in dengue surveillance data. In D10, a case without any geocoded dengue case information, the authors employed a GIS-based analytical hierarchy process to predict dengue risk zones using a set of environmental and socio-economic variables. Though the outcome of D10 was unvalidated, the map was still helpful for policymakers by providing potential hot spots for dengue outbreaks. 

In addition, the application purposes of predictive maps determine the modelling approach as well. In the case of a fine resolution map being needed to spatially display the hot spots of DF burden to support public health actions, the GWR and machine learning methods were often a priority, due to their specialties in predicting spatial autocorrelation and dealing with multifactorial datasets. However, in the case of developing an EWS for the local government to support health system preparedness for DF outbreaks, Bayesian models were commonly adopted, as they allowed for optimal resource allocation and steering of interventions in space and time. 

### 4.5. Improvement Suggestions

First, even though it was one of the weaknesses mentioned in Louis et al. [12], the impacts of the serology profile and virus genetic diversity on the spatial pattern of DF risk were still ignored in all selected studies. Both the host immunity and serotyping of the dengue virus play an important role in diffusion models [67]. To integrate dengue serotyping data for detecting spatial patterns, cohort studies on both entomological and epidemical dengue data may be necessary to understand the transmission dynamics and determinants within urban environments [68].

Secondly, the datasets for socio-economic factors applied in selected studies were limited and had low spatio-temporal resolutions. Both D14 and D18 reported a more significant impact of socio-economic factors on dengue transmission than that of climatic and built environmental aspects, due to the huge spatial heterogeneity among socio-economic groups in high density cities. Restricted by the number and range of samplings, a questionnaire-based survey on socio-economic status cannot meet the requirement of mapping DF risk for an entire urban environment [69]. A combination of data mining and crowdsourcing may be an effective solution to support socio-economic analysis in future studies [70].

Lastly, almost all selected papers adopted a distance-based assumption on spatial connectivity for generating predictive maps, which significantly simplified the transmission process, but may have missed some heterogeneity in connectivity in urban spaces [23]. Only D3 applied cellphone data to track individual mobility to describe spatial connectivity in a city and found that long-distance connections may also be important for disease transmission, especially for regions under a high risk of importation. However, cellphone data are commonly unavailable for major regions due to ethical considerations. Therefore, retrieving data from social media [71] or public transportation [72] may be an adequate data source to demonstrate human commuting behaviors for future research. 

### 4.6. Limitations

Some limitations of this research included: (1) The restriction of language to English-only papers. (2) This review solely focused on dengue-related risk, without considering other, potentially similar vector-borne diseases, such as Zika, West Nile, etc. (3) Some details about spatial models, such as integration effects in Bayesian models and distributions, were not collected and compared in this review as they had already been investigated by previous studies [11,23].

## 5. Conclusions

This review focused on the modelling approaches and influential factors used to develop predictive maps for DF risk in urban settings. Both the data sources and mapping techniques were significantly improved compared with Louis et al. [12]. Big data and emerging data-mining techniques, such as cellphone and street view images, have been innovatively introduced for mapping DF risk in urban settings, as they can offer more details about human mobility behavior and micro-habitats for mosquito breeding. There was clear trend preferring the use of Bayesian approaches and machine learning algorithms for modeling dengue-related risk in recent years. Temperature and rainfall-related climatic factors, land use and land cover information, and demographic data were the most significant predictors for generating the DF risk map. The impacts of influential factors summarized in our study can be a reference for future studies. So far, the analysis pattern for developing a predictive map of DF risk is beginning to take shape. Mapping methodologies are largely determined by data availability and the purpose of application. Last but not least, to enhance current abilities in detecting transmission dynamics and heterogenous spatial distribution of DF risk in urban settings, future models need to consider integrating information about host serological profiles, promoting data availability on socio-economic factors, and employing adequate spatial connectivity assumptions in the process of modeling. Having robust predictive maps on DF risk will be crucial for supporting local public health units in controlling the spread of the dengue virus, especially in the context of global warming that expands suitable niches for mosquitoes and raises the risk of dengue transmission.

## Figures and Tables

**Figure 1 ijerph-19-15265-f001:**
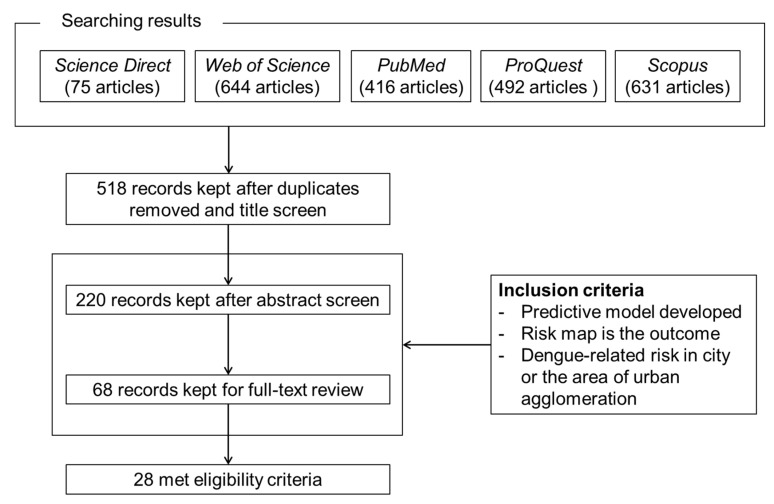
Flow diagram of article selection and inclusion process.

**Figure 2 ijerph-19-15265-f002:**
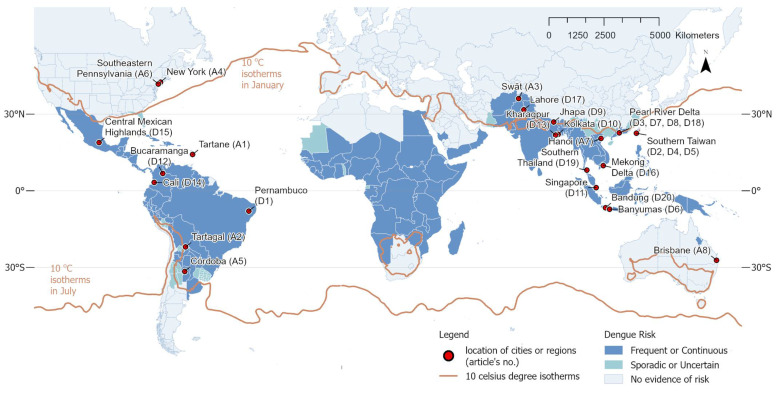
Location of selected studies overlaid with the map of the global dengue burden. The levels of dengue risk are based on reports from the Centers for Disease Control and Prevention, National Center for Emerging and Zoonotic Infectious Diseases, and Division of Vector-Borne Diseases [52]. “Frequent or Continuous” risk means that either frequent outbreaks occur, or transmission is ongoing. “Sporadic or Uncertain” risk means that the risk varies and is unpredictable, and that country-level data is not available. The 10 °C isotherms in the winter season were generated using the mean temperature (2 m height, world) data from January and July in the year 2018 retrieved from the ERA-Interim.

**Figure 3 ijerph-19-15265-f003:**
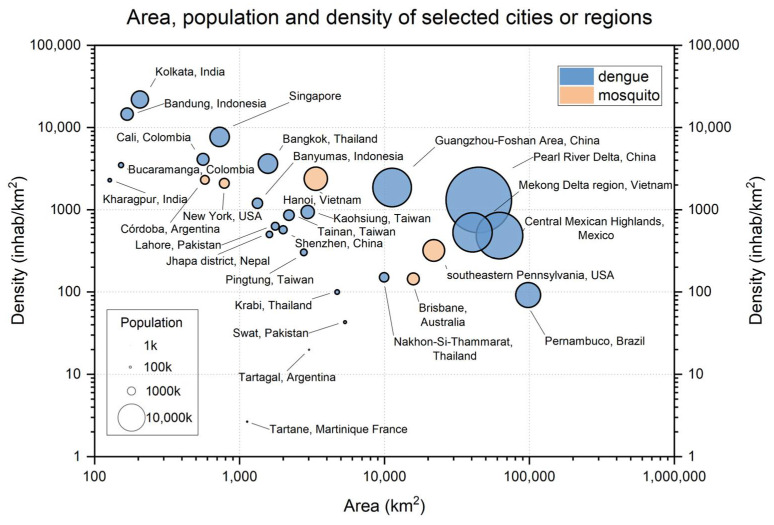
General information about the study area in selected references.

**Figure 4 ijerph-19-15265-f004:**
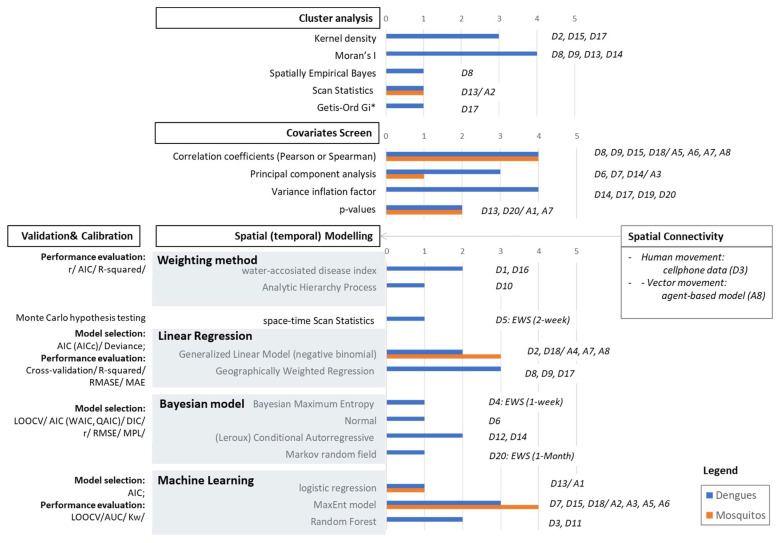
The modeling approaches of cluster analysis, covariates screen, spatial or spatio-temporal modeling, spatial connectivity, and validation methods in selected studies.

**Table 1 ijerph-19-15265-t001:** Keywords for the set of four-level querying.

Levels	Key Words for Querying
1	‘dengue’ OR ‘dengue fever’
2	‘risk’ OR ‘vulnerability’ OR ‘hot spot’
3	‘map*’ OR ‘model*’
4	‘spatial’ OR ‘spatiotemporal’ OR ‘distribution’

**Table 2 ijerph-19-15265-t002:** List of publications selected for critical review.

No.	References	No.	References	No.	References
D1	Dickin and Schuster-Wallace [24]	D11	Ong et al. [26]	A1	Machault et al. [25]
D2	Wen et al. [27]	D12	Martinez-Bello et al. [28]	A2	Espinosa et al. [29]
D3	Mao et al. [30]	D13	Ghosh et al. [31]	A3	Fatima et al. [32]
D4	Yu et al. [33]	D14	Desjardins et al. [34]	A4	Little et al. [35]
D5	Chen et al. [36]	D15	Ordonez-Sierra et al. [37]	A5	Estallo et al. [38]
D6	Wijayanti et al. [39]	D16	Pham et al. [40]	A6	Wiese et al. [41]
D7	Li et al. [42]	D17	Naqvi et al. [43]	A7	Ha et al. [44]
D8	Ren et al. [45]	D18	Wu et al. [46]	A8	Trewin et al. [47]
D9	Acharya et al. [48]	D19	Yin et al. [49]		
D10	Ajim Ali and Ahmad [50]	D20	Jaya and Folmer [51]		

**Table 3 ijerph-19-15265-t003:** Effect of climatic variables on dengue-related risk.

Variables	Positive	Negative	Nonlinear	Unknown
Air temperature	D20 (monthly); A3 (monthly), A4(monthly),	D14 * (weekly),	D7 (warmest month); D18 (seasonally), A6(seasonally)	D3, D15
Daily temperature range	D14 * (weekly)	A3 (monthly),		
Cool days (<18 °C)		D14 * (weekly)		
Warm days (>32 °C)	D14 * (weekly)			
LST	D8, D9, D10 (>25 °C)	D12, D13, D17, D19		
nLST	D6 (<20 °C)			
Relative humidity	D8 (yearly); A1 (daily), A4 (monthly)			
Water vapor pressure	D20 (monthly);			
Precipitation	D8 (yearly), D19 (seasonally), D20 (monthly); A1 (daily), A4 (monthly), A6 (seasonally), A8 (monthly)	D18 (monthly), D14 * (weekly)	D7 (warmest month); A3 (monthly),	D3, D15
Solar radiation	D20 (monthly)			

* Lagged weather variables.

**Table 4 ijerph-19-15265-t004:** Effect of built environmental variables on dengue-related risk.

Types	Variables	Positive	Negative	Nonlinear	Unknown
Land Cover	Built-up/ impervious area	D7, D8 *, D17, D18 *;			
		A2, A5, A6	A1 (asphalt)	A6 *	
	NDVI (EVI)	D17; A5, A7	D8 *, D9, D11, D13	D7 (warm season); A6 *	D15
	Trees density/canopy	D14		A6 *	
	NDWI	A2	A4 *	A6 *	
Land Use	Residential use	D11, D13; A4 *			
	Open space use		A4 *		
	Greenings	A3, A4		A1, A6 *	
	Vacant use	A2			
	Unplanned area	D13; A4			
Morphology	Property size		A8		
Landscape	Proximity to parks/managed vegetation			D3 *	
	Proximity to water bodies/rivers	D13; A5	D14;	D3 *	
Road	Road density	D8 *, D11, D7, D18		D3 *	
	Road length		A8		
Establishment	Proximity to specific establishment	D14 (tires and plant nurseries); A5 (cemeteries, tires)	D6 (hospital)	D2 (species), D3 * (workplaces)	
Infrastructure	Container density			D19 (types)	
	Water tanks	A8			
	Drain network	D13			
Topography	Flow accumulation		A6 *		
	Elevation	D13		D3; A3	A6 *
	Slope				A6 *

* *Aedes albopictus*.

**Table 5 ijerph-19-15265-t005:** Effect of socio-economic variables on dengue-related risk.

Types	Variables	Positive	Negative	Nonlinear	Unknown
Population	Population density	D8 *, D19, D18 *; A7, A8	D14	D7; A6 *	A3
	Household density (households/100/km^2^)	D11; A5, A6 *			
Development	GDP	D8 *, D18 *			
Demography	Age > 65/60	D14			
	Age < 14/15/school	D14			
	Low education		D6, D14		A6 *
	Low income		D14		A6 *
	Rate of unemployment	D14	D6		
Living conditions	Infection history	D11			
	Vacant housing			A6 *	
	Neighborhood quality		A5		A6 *
	Without piped water	A5			

* *Aedes albopictus*.

## Data Availability

Not applicable.

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
