# Peer review of "A Systematic Review on Modeling Methods and Influential Factors for Mapping Dengue-Related Risk in Urban Settings"

_ijerph, 2022, doi:10.3390/ijerph192215265_

Round 1
Reviewer 1 Report
The authors summarized the current findings about modeling methods and influential factors for the risk of dengue fever infections. The paper is organized well and draws a high impact in infectious disease control in urban settings. Nevertheless, I have some major comments:
1. "A review" or "a systematic review"? It should be defined well as STROBE checklist is required.
2. How to define "mapping"? May mapping in global scale be included? or studies only mapping within an area be included?
3. Why only modelling studies? How about cohort?
4. If there is a study only drew a risk prediction using various models without a mapping, would that be included? why risk mapping is a criteria should be elaborated.
5. Different studies had various sources of data available especially for the built environmental variables. Would the summarization of influential factors be useful when there is such a limitation of data heterogeneity?
6. Dengue fever is some urban settings may be driven by the import cases and this may have a great difference compared to the areas having dengue fever as an endemic. How this be accounted in the study?
7. Most importantly, the authors lack highlighting the important of what they summarized. What can be suggested based on the findings? The conclusion is TOO general that can be applied to every investigations.
Author Response
The authors summarized the current findings about modeling methods and influential factors for the risk of dengue fever infections. The paper is organized well and draws a high impact in infectious disease control in urban settings. Nevertheless, I have some major comments:
- "A review" or "a systematic review"? It should be defined well as STROBE checklist is required.
Response: Thank you for your comments. In this study, we followed the method and structure of systematic review. We revised the title into “A Systematic Review on Modeling Methods and Influential Factors for Mapping Dengue-Related Risk in Urban Settings” according to your suggestion. In addition, we think the checklist you mentioned here should be PRISMA checklist (https://prisma-statement.org/prismastatement/Checklist.aspx), since the STROBE checklist serves for observation study. We added the checklist in the part of supplementary materials.
- How to define "mapping"? May mapping in global scale be included? or studies only mapping within an area be included?
Response: We added the definition of mapping in the Introduction section. Please refer to Line 59-69. The most significant application of mapping dengue-related risk is the visualized spatial pattern and determinants from the predictive models, which are valuable for local governments and stakeholders to take actions on controlling dengue transmission.
Regarding the scale issue, we added a new paragraph. Please refer to Line 81-93 (marked with yellow). In general, we didn’t include the global scale but only focused the mapping developed for urban area. There are three reasons: 1) distinct environmental and socioeconomic differences between urban and rural area; 2) coarse and uncertain results from mapping of dengue endemicity on the global scale; 3) the limitation of model developed with site-specific data. In this manuscript, we only focused on reviewing the studies which developed predictive mappings for dengue-related risk for a city or urban aggregated regions.
- Why only modelling studies? How about cohort?
Responses: Thank you for proposing this interesting question. Generally, the data sampling methods for mapping study and cohort study are fundamentally different. The former is latitudinal or horizontal study, which benefit for detecting the spatial pattern within a region, while the latter is longitudinal study for investigating more detailed and intrinsic factors relating to the disease transmission. As the cohort studies commonly are incapable to generate the map or predict the area without sampling data, we only focused on the studies with a predictive map as outcomes. However, it is very important to consider the cohort study in our future research on dengue fever risk in urban settings.
- If there is a study only drew a risk prediction using various models without a mapping, would that be included? why risk mapping is a criteria should be elaborated.
Response: The reason for only focusing on mapping studies is that the resultant maps not only intuitively illustrate the spatial or spatiotemporal pattern of the dengue-related risk visually, but also propose determinants as the out-comes of spatial modelling. Taking the advantage of visualization and prediction, mapping dengue-related risk is valuable for local governments and stakeholders to detect the location of hot spots and take actions for controlling disease transmission. Therefore, we only included the studies with mapping. The other studies drew a risk prediction might be capable for mapping as well, but we can not identify directly by our reviews. We supplied the elaboration on this issue in this revised manuscript. Please see Line 59-69 (marked with yellow).
- Different studies had various sources of data available especially for the built environmental variables. Would the summarization of influential factors be useful when there is such a limitation of data heterogeneity?
Response: Thank you for your comments. Indeed, the data availability varies with locations and cities. There are two reasons for summarizing the influential factors in our study: 1) for interpreting what kind of data can be applied for mapping dengue-related risk in urban settings; 2) for illustrating how the influential factors impacts the dengue-related risk. We hope our study findings can be a reference for future study on the aspect of collecting data and checking the reliability of the predictive models. We added this in the Conclusion section.
- Dengue fever is some urban settings may be driven by the import cases and this may have a great difference compared to the areas having dengue fever as an endemic. How this be accounted in the study?
Reponse: In our study, we separated the entomological and epidemical dengue fever risk mapping into two groups. Generally, according to our review, the study on non-endemic region prefers employing the dengue vector surveillance data for assessing the entomological dengue fever risk, due to the lack of dengue cases for developing the predictive map. Some studies on the area were affected by the import cases significantly in specific years, so that the non-endemic area might be transformed as endemic area. Thus, we didn’t take this issue into account in our study, since the underlying determinants behind the dengue fever risk were similar among these cases.
- Most importantly, the authors lack highlighting the important of what they summarized. What can be suggested based on the findings? The conclusion is TOO general that can be applied to every investigations.
Response: Thank you for pointing out this issue. The conclusion part has been revised accordingly. Our key findings are highlighted. Please refer to the revised version (marked with yellow).

Reviewer 2 Report
The manuscript provides a comprehensive review of methods and determinants of mapping dengue-related risk in urban settings, focusing on the literature after 2013. The paper is well structured and presents how dengue is mapped clearly. But there are several major issues.
1. As mentioned in paged 12, “Louis, et al. is the only review study about the available mapping tools basing on 68 papers published until 2014”. The difference in the findings between this paper and Louis et al. (2014) should be intensely discussed, which helps to highlight the innovation and significance of the research.
2. Also, in comparing the findings of the manuscript with that of Louis et al., it’s important to briefly discuss the change and its reason in the research focus, research data, method, and research area. This enables readers to get a full picture of the research progress in the field and simultaneously underlines the significance of the review on the literature after 2014.
3. Additionally, regional heterogeneity is an important reason for policy makers to adopt different actions in coping with dengue-related risk in different areas. Thus, a deeper discussion of why the scholars focus on urban settings in several particular locations and how they conduct the research differently is necessary.
4. The paper presents the research progress clearly. But the critical thinking is also important in the review to inspire further research, which should be strengthened in the discussion part.
Author Response
The manuscript provides a comprehensive review of methods and determinants of mapping dengue-related risk in urban settings, focusing on the literature after 2013. The paper is well structured and presents how dengue is mapped clearly. But there are several major issues.
- As mentioned in paged 12, “Louis, et al. is the only review study about the available mapping tools basing on 68 papers published until 2014”. The difference in the findings between this paper and Louis et al. (2014) should be intensely discussed, which helps to highlight the innovation and significance of the research.
Response: Thank you for pointing out this issue. The discussion part has been revised accordingly. The difference in the findings between our study and Louis et al. (2014) are discussed in the aspects of study area, effective predictors, and modeling techniques. Please refer to the revised version (Line 330)
- Also, in comparing the findings of the manuscript with that of Louis et al., it’s important to briefly discuss the change and its reason in the research focus, research data, method, and research area. This enables readers to get a full picture of the research progress in the field and simultaneously underlines the significance of the review on the literature after 2014.
Response: Thank you for your suggestions. The difference is briefly summarized at the both the beginning of Discussion part and Conclusion part. Please refer to Line 331-334, and Line 519.
- Additionally, regional heterogeneity is an important reason for policy makers to adopt different actions in coping with dengue-related risk in different areas. Thus, a deeper discussion of why the scholars focus on urban settings in several particular locations and how they conduct the research differently is necessary.
Response: Thank you for your comments. The reason for why we only focused on urban settings in several particular locations is that we wish the outcomes of the study about the modeling method and influential factors can be references for other regions to develop their own risk map for preventing dengue transmission. We added an elaboration in the Introduction part. Please see the Line 81-93.
Regarding the issue about how they conducted the research differently, we also added a section for discussing the mapping methodology design to interpret the reason behind of how they conduct the research differently. Please see the Line 461-481.
- The paper presents the research progress clearly. But the critical thinking is also important in the review to inspire further research, which should be strengthened in the discussion part.
Response: Thank you for pointing out this. The whole section of Discussion has been revised. We also added a section (4.5) for interpreting the suggestions on improving the predictive maps for future studies.

Reviewer 3 Report
This manuscript briefly reviewed and summarized modeling methods and influential factors for mapping dengue fever risk in urban settings since 2014. It merits publication if the authors can supplement calibration or validation of the modeling methods.
Author Response
This manuscript briefly reviewed and summarized modeling methods and influential factors for mapping dengue fever risk in urban settings since 2014. It merits publication if the authors can supplement calibration or validation of the modeling methods.
Response: Thank you for your comments. We added a section (3.3.4) in Results and a part (4.3.2) in Discussion part on the calibration or validation of those modeling methods in our reviewed studies.

Round 2
Reviewer 2 Report
The authors have addressed all of my concerns in the revised manuscript.